# Effects of Combined HIIT and Stroop on Strength Manifestations, Serve Speed and Accuracy in Recreational Tennis Players

**Juan Pedro Fuentes-García** [1], **Jesús Díaz-García** [1], **Miguel Ángel López-Gajardo** [1,*] and **Vicente Javier Clemente-Suarez** [2,3]

1   Faculty of Sport Sciences, University of Extremadura, 10003 Cáceres, Spain; jpfuent@unex.es (J.P.F.-G.); jdiaz@unex.es (J.D.-G.)
2   Faculty of Sport Sciences, University European of Madrid, 28670 Villaviciosa de Odón, Spain; vctxente@yahoo.es
3   Grupo de Investigación en Cultura, Educación y Sociedad, Universidad de la Costa, Barranquilla 080002, Colombia
*   Correspondence: malopezgajardo@unex.es

**Abstract:** Background: The importance of the serve in tennis players' performance is well known but no previous studies have analyzed the effects of both physical and cognitive fatigue on the speed and accuracy of the serve. This study analyzed the effect of a High-Intensity Interval Training (HIIT) with and without cognitive load on serve speed and accuracy, spirometry, and strength manifestation. Methods: 32 recreational players (25 men and 7 women; aged $21.40 \pm 1.52$ years) performed a HIIT and a HIIT with a Stroop in recovery phases before performing a series of tennis services. Speed and accuracy of the services, spirometry, and strength manifestations were registered. Results: The main findings of the study showed that strength manifestations and spirometry were not affected by either protocol. A decrease in serve speed was observed in both protocols ($p < 0.001$) but service accuracy did not show impairments ($p = 0.66$). Conclusion: A combination of physical and mental fatigue may decrease serve speed but will not affect strength manifestations or spirometry negatively. These results could be caused by a response of the central nervous system to maintain the accuracy of the ball in presence of fatigue.

**Keywords:** physical fatigue; cognitive fatigue; mental fatigue; tennis service performance

## 1. Introduction

Tennis is characterized by powerful and accurate actions. The result of a match is greatly influenced by the "breaks", or broken services, the percentages of successes-errors, double faults, or errors in returns due to services with high speed and accuracy [1]. Therefore, the importance of the serve in tennis performance is highlighted [2].

Considering the relevance of service, several studies have analyzed speed [3,4] or accuracy [5,6] of the serve from different perspectives. Although speed and accuracy have been shown to be related, few studies have simultaneously analyzed the speed and accuracy of the serve [7,8], or even the relationship between the speed and the accuracy of the serve, taking into account the effect of a physical load on these variables. In this regard, Terraza et al. [9] examined the post-activation potentiation effect on serve speed and accuracy in young competition tennis players using high-intensity exercises. They did not find any significant effect of upper and lower body heavy-load resistance exercises on serve accuracy in young competition tennis players. High-Intensity Interval Training (HIIT) is a commonly used method to train physical aspects in tennis based on the similar metabolic demands between this type of training and tennis matches, however, the effects of HIIT on a relevant hit such as the tennis service have not been widely studied, despite

the fact that this information may allow coaches to understand tennis service performance during matches and training in presence of physical fatigue.

Tennis also implies mental demands [10]. Van Cutsem et al. [11] stated that mental fatigue in sports situations is caused by cognitive load (i.e., tactical decisions) and emotions (i.e., remembering previous failures), which can impair sports performance. Specifically, these authors indicated that mental fatigue produced a decline in endurance performance (e.g., power output/velocity or increased completion time), meanwhile, Habay et al. [12] highlighted that mental fatigue may impair specific sport psychomotor performance. Previous studies have supported these ideas, reporting negative effects of mental fatigue on racket sports [13,14]. These impairments caused by mental fatigue have been associated with an increase in the ratio of perceived exertion [11], poor cognitive performance, or a response of the central nervous system to maintain the accuracy in the presence of fatigue [13–15], among others. Although motivation may influence the negative effects of mental fatigue on performance [16,17], a negative effect of mental or cognitive fatigue on tennis service may also be present but it has not yet been tested.

Based on the above information, although tennis service may be negatively affected by physical and cognitive fatigue, there is a lack of research that studies the effects of both types of fatigue on serve speed and accuracy. Physical fatigue is normally associated with physically fatiguing tasks, and it could be identified by impairments in traditional physiological systems analyzed as heart rate output, blood lactate, or oxygen uptake; meanwhile, mental fatigue is a psychobiological state caused by prolonged cognitive demanding tasks, where it has been demonstrated that contrary to physical fatigue, the role of the brain is evident in this type of fatigue [18]. Tennis service is a complex psychomotor action that could be influenced by physical fatigue (i.e., capacity to produce force) or mental fatigue (i.e., synchronization). During tennis matches, both types of fatigue are synergizing, therefore, the information provided by the study of a combination of physical and mental fatigue could improve the training programs of coaches, synergizing physical, mental, and technical aims [10]. In this regard, we consider it interesting to study the effects of HIIT both on accuracy and serve speed. It is also interesting to include a HIIT with cognitive load because it represents an additional impact on the accuracy and speed factors. Therefore, the present study aimed to analyze the effect of introducing a cognitive task in a specific tennis HIIT session on serve speed and accuracy, spirometry, and strength manifestations in tennis players.

The initial hypothesis was that HIIT would decrease strength manifestations and serve speed and accuracy. Also, we hypothesized that the implementation of a cognitive task would decrease strength manifestations and spirometry variables as a fatigue marker and also decrease participants' service ball speed and accuracy.

## 2. Materials and Methods

### 2.1. Participants

According to the purpose of the study and the measures designed, an analysis of the sample size needed was performed with G*Power. This analysis indicated that a minimum sample size from 23 to 26 players was needed.

A total of 36 volunteer recreational players (27 men and 9 women) were enrolled in the study. However, 32 participants (25 men and 7 women), with an average age of 21.40 (SD = 1.52) years and an average experience in tennis practice of 0.84 years (0.80) with 3.26 (0.78) hours of weekly tennis training, completed the procedure. The players, 30 right-handed and 2 left-handed, voluntarily participated in the study, providing their written signed consent. Exclusion criteria included injuries of any type that had occurred in the four months before the study. Participants were encouraged to avoid the consumption of caffeine for at least 1 h prior to participating in the study. Also, they were encouraged to avoid the consumption of creatine during the study.

This research complied with the Helsinki declarations (revised in Brazil, 2013), on human research and was approved by the University Ethics Committee (CIPI/18/093).

## 2.2. Procedures and Materials

Tennis players had to perform seven services in three different situations: (1) basal condition, without application of HIIT or cognitive load, (2) after performing a HIIT without cognitive load and (3) after performing a HIIT with cognitive load. Participants were encouraged to perform each service at the maximum power, trying to send the ball to the maximum score area (9 points). This service expression is closer to tennis service than the second service, due to the second service implying another type of effect (i.e., topspin) that would influence the results of the study, meanwhile, the purpose of the present study was to test the effects of both fatigue protocols on serve speed-accuracy index. An adaptation of the criteria established by Menayo et al. [1] was used to distribute the track areas and determine their dimensions (Figure 1). While these authors examined other aspects of tennis service, we tested the speed and accuracy of participants when they tried to obtain 9 points, therefore, an adaptation of the punctuations as per the objective of the task had to be performed. A researcher controlled the process to ensure all tennis services were performed according to the International Tennis Federation Rules. When one service was not within the specified zones, the punctuation was 0, and if a tennis service touched the net and passed to obtain a valid service, the International Rules were checked (i.e., this service was repeated). No faults or repeated services were reported by this researcher.

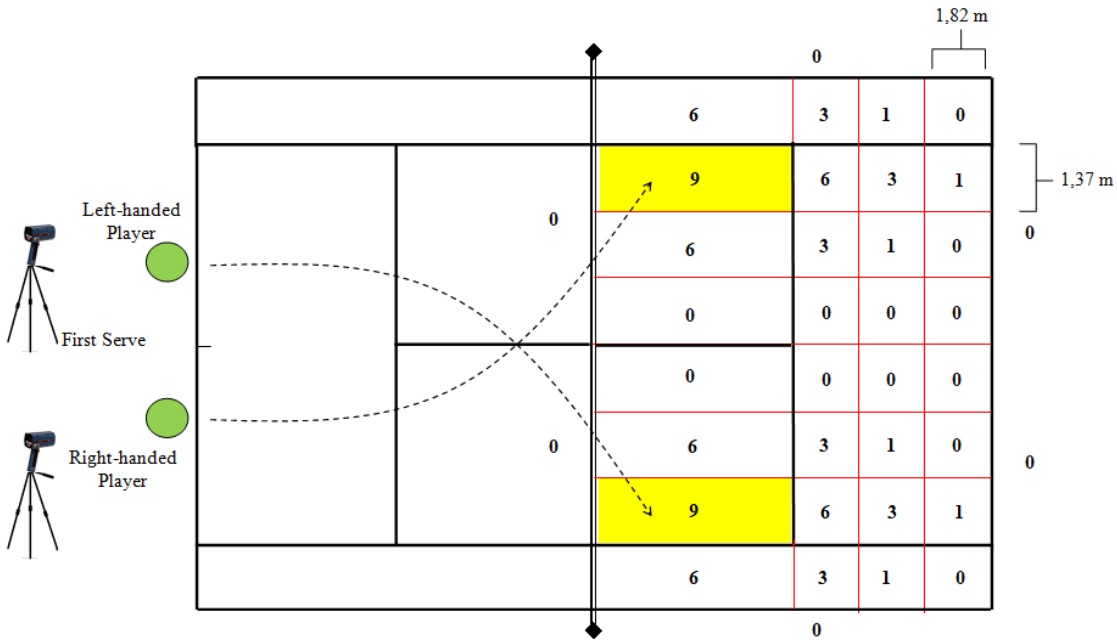

**Figure 1.** Tennis service test explanation.

To avoid the influence of previous information during the procedure, players were alone while they performed the serves. Each of the protocols (basal, HIIT, and HIIT with cognitive load) was performed with a 48-h difference between them. The participants performed in randomized counterbalanced order to avoid the effects of one protocol on another. Moreover, all tennis players were informed about the zonal distribution of the tennis court, the score assigned to each zone, their placement in the service zone, and the number of services to be performed. The three different protocols were conducted in an indoor tennis court with the dimensions regulated by the International Tennis Federation.

A modified version of a previous HIIT structure was used [19]. The HIIT implies 12 repetitions of 30 s of push-ups, squats, and lateral displacements with another 30 s of rest. In the HIIT with cognitive load, 30 s of incongruent Stroop were performed during the rest. The validity of this type of short- task to induce mental fatigue has been

previously summarized in Van Cutsem and Marcora [20]. Taking into account these authors, a researcher controlled the engagement of the participants with these tasks. For the tennis service actions, each player previously performed five minutes of individual and service-specific warm-ups. One researcher provided the balls to the tennis players to prevent the participants from moving from the initial position required for all services. This researcher indicated every four seconds when the next serve must be performed. Another researcher recorded the ball speed indicated in the radar. Lastly, another researcher recorded the score obtained in each service based on the area the ball touched.

### 2.3. Outcomes

Following the protocol of a previous study [21], the main outcomes of the present study were:

1.  Serve ball speed (hitting power). A radar (Stalker Radar Sport 2), with $\pm 1$ km/h accuracy, was employed to measure the ball's speed during the services. A tripod was placed 1.5 m behind the centerline. The radar was orientated from the shot spot to the bullseye to avoid mistakes related to the angle formed by the radar and the ball's trajectory, following the protocol of previous studies [21].
2.  Lower body muscular power was evaluated through horizontal countermovement jump tests. Participants performed a standardized warm-up consisting of $2 \times 10$ vertical jumps with 30 s recovery (previously) and then, they performed two maximal horizontal countermovement jumps with hands placed on the waist, recovering for 15 s between them, as previously reported [22], and the best attempt was used for the statistical analysis.
3.  Spirometry: Forced Vital Capacity (FVC), volume exhaled at the end of the first second of forced expiration (FEV1) and the peak expiratory flow (PEF) using a QM-SP100 (Quirumed, Spain) spirometer during a maximum inhale-exhale cycle following previous protocols [23].
4.  Isometric hand strength (IHS) with a grip dynamometer (Takei Kiki Koyo, Japan). Participants were in anatomical position with the dominant arm extended. From that position, they were asked to perform two maximum contractions of the hand from 3 s with 3 s of recovery between them. The highest value was used for the research [24].
5.  Skin temperature (ST) was measured with a digital infra-red thermometer (Temp Touch; Xilas Medical, San Antonio, TX). With the subject in anatomical position, the area of the right temple was cleaned with sterile gauze and then the temperature of the area was taken with an infrared thermometer [25].

For the basal sample, participants conducted the serve ball test firstly, after this a warm-up consisting of $2 \times 10$ vertical jumps as previous research [26], and then the ST, horizontal jump, spirometry, IHS tests, and skin temperature with the recovery were described. The post HIIT collection followed the same procedure excluding the warm-up.

### 2.4. Statistical Analysis

Data were analyzed using the Statistical Package for the Social Sciences (SPSS) version 21.0 (SPSS Inc., Chicago, Ill., USA). The Shapiro–Wilk test was conducted to study the normality of the data. In this regard, non-parametric statistical tests were employed. The Friedman Test Effect Size (Kendall's W Value) was performed to explore the impact of the three protocols on ball speed and accuracy. It is classified as: 0.5 as a large effect, 0.3 as a medium effect, and 0.1 as a small effect [27]. Wilcoxon signed-rank tests were performed to conduct pairwise comparisons. The *p*-values were corrected using Bonferroni adjustment for multiple comparisons to avoid Type I errors.

## 3. Results

Results showed no significant differences in the ball speed between the different conditions evaluated ($p = 0.66$) although significantly lower accuracy scores were observed in presence of HIIT and cognitive load (Table 1).

**Table 1.** Effects of HIIT without and with Stroop on the ball speed and accuracy in the tennis service.

| Test | Baseline Mean (SD) | HIIT without Cognitive Load Mean (SD) | HIIT with Cognitive Load Mean (SD) | Chi-Squared | *p* * | Effect Size |
|---|---|---|---|---|---|---|
| Ball speed [km/h] | 98.24 (23.22) | 102.92 (23.29) | 99.76 (20.35) | 0.813 | 0.666 | 0.05 |
| Accuracy average score [points] | 4.88 (1.20) | 3.88 (1.37) | 3.60 (1.36) | 13.95 | 0.001 | 0.87 |

* *p*-value obtained from the Friedman test.

The Wilcoxon signed-rank test showed significant differences in the ball speed between baseline and HIIT without cognitive load ($p = 0.002$) and HIIIT with cognitive load ($p = 0.015$; see Figure 2). However, no significant differences were found in the average accuracy score between protocols (see Figure 3). These results suggest that physical fatigue and a combination of cognitive and physical fatigue mainly affect the speed of tennis service negatively, without significant changes in the accuracy.

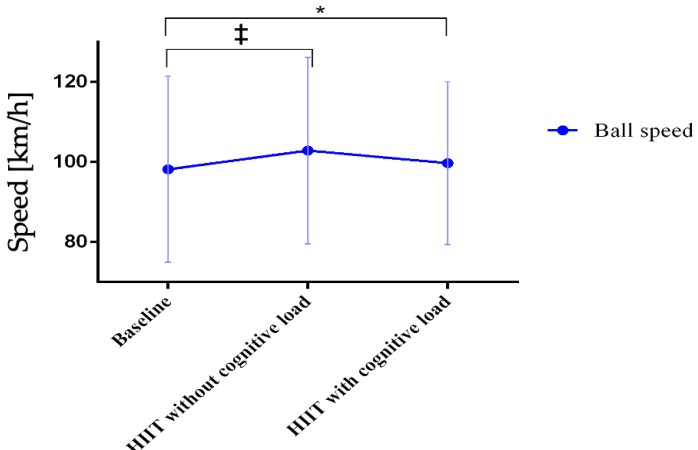

**Figure 2.** Comparison between three conditions in ball speed. * Ball speed after HIIT exercises with cognitive load is significantly greater when compared with ball speed at baseline ($p < 0.05$). ‡ Ball speed after HIIT exercises without cognitive load is significantly greater when compared with ball speed at baseline ($p < 0.05$).

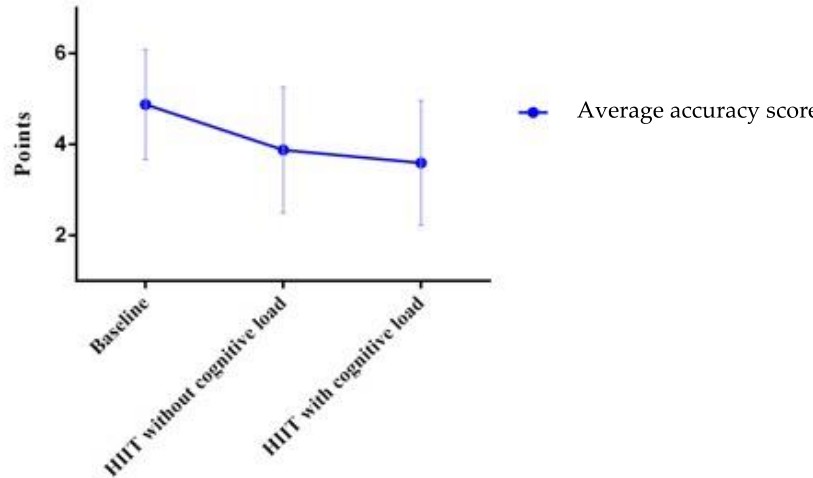

**Figure 3.** Comparison between the three different conditions in the average accuracy score.

Finally, in the other variables analyzed, significant differences were only found in Body Temperature (°C); the temperature was lower at the Post (2) than at the Pre (1) of the HIIT without cognitive load and then, the temperature was lower at the Post (2) than at the Pre (1) of the HIIT, both without and with cognitive load (Table 2). These results suggest that neither condition described impaired strength manifestations or spirometry variables.

**Table 2.** Effects of HIIT without and with Stroop in the strength and body temperature measures.

| Variables | HIIT without Cognitive Load Mean (SD) | | HIIT with Cognitive Load Mean (SD) | | *F* | *p* | η2 | Moment Comparison |
| --- | --- | --- | --- | --- | --- | --- | --- | --- |
| | Pre (1) | Post (2) | Pre (3) | Post (4) | | | | |
| Horizontal Jump Test [cm] | 167.4 ± 29.0 | 169.9 ± 24.6 | 170.3 ± 26.9 | 170.1 ± 28.8 | 0.448 | 0.720 | 0.044 | |
| FVC | 4.9 ± 1.3 | 4.8 ± 1.4 | 4.9 ± 1.3 | 4.6 ± 0.7 | 0.994 | 0.409 | 0.093 | |
| FEV1 | 3.7 ± 0.9 | 3.6 ± 0.7 | 3.9 ± 0.7 | 3.7 ± 0.9 | 1.417 | 0.258 | 0.128 | |
| PEF | 8.1 ± 2.0 | 7.8 ± 2.23 | 8.5 ± 2.0 | 7.8 ± 2.2 | 2.526 | 0.077 | 0.207 | |
| Isometric Hand strength [N] | 41.0 ± 10.9 | 40.5 ± 9.3 | 40.4 ± 10.0 | 41.5 ± 9.9 | 0.552 | 0.651 | 0.054 | |
| Skin Temperature [°C] | 32.0 ± 1.0 | 30.9 ± 1.6 | 32.0 ± 1.0 | 31.6 ± 1.5 | 5.312 | 0.005 | 0.355 | 1 > 2; 3 > 2 |

## 4. Discussion

The main purpose of this study was to analyze the effects of adding a cognitive task in a specific tennis HIIT session on serve speed and accuracy, spirometry, and strength manifestations in tennis players. The results of the present study showed that both conditions significantly decrease tennis serve speed, whereas the accuracy of the service, spirometry, and strength manifestations was not significantly affected by either of these protocols.

To our knowledge, this is the first study that analyzed the effects of HIIT on serve speed and accuracy. We hypothesized that HIIT would decrease the speed and accuracy of the tennis service. The results of this study suggest that physical fatigue induced by a HIIT decreases the serve speed, without significantly changing service accuracy, strength manifestations, or spirometry. Therefore, we can only partially confirm this hypothesis. A previous study has analyzed the effects of physical fatigue on service, without observing significant effects [9]. In contrast, significant positive effects on service have been reported after tennis matches implementing physical recovery protocols [28]. This lack of agreement could be explained by the type of exercise used to induce fatigue. The findings of Suárez-Rodríguez and Del Valle [29] support this explanation, as they reported different effects between HIIT and interval training in the performance of different tennis shots, with significant major fatigue indicators after a HIIT protocol. Therefore, we suggest that the specific fatigue induced by performing a HIIT significantly decreased the speed of the service.

This is also the first study that has analyzed the implementation of a cognitive fatigue protocol added to HIIT in these variables. Although we hypothesized that a combination of cognitive load added to HIIT would decrease both speed and accuracy of tennis service, only speed was impaired. Therefore, we can only partially confirm this hypothesis, suggesting that a combination of cognitive and HIIT fatigue decreases the speed of tennis service. A significant decrease in the speed of the ball when facing a mental fatigue-induced protocol (i.e., Stroop) has been previously tested in racket sports [13]. Although more studies are necessary to test this information, it seems that a combination of both types of fatigue

may increase the impairments caused by fatigue in service. Higher impairments in sports performance caused by dual tasks have been previously reported [30].

We also hypothesized that both protocols would decrease spirometry and strength manifestations. On the contrary, these variables did not show significant impairments after HIIT. These results suggest that these protocols did not impair the traditional systems associated with sports performance, but certain neural mechanisms and the central nervous system may be affected by these types of tasks [31]. Previous studies have reported that the negative effects of cognitive fatigue on physical fatigue are not caused by impairments in the traditional physiological systems [18]. In fact, cognitive fatigue seems to impair physical performance by increasing the subjective ratio of perceived exertion [11,32], where the role of adenosine seems a notable candidate [31]. This may explain why cognitive fatigue added to HIIT did not significantly impair strength manifestations or spirometry.

This previous information also suggests that the decreases reported in the speed of the serve were not caused by an impairment in these traditional physiological systems. Thus, the decrease in serve speed may be a response of the central nervous system to maintain accuracy in presence of fatigue. This explanation was provided in a previous study performed with table tennis players [14], as these authors reported a significant decrease in the speed of the ball when facing a mental fatigue-induced protocol (i.e., Stroop), without changes in the accuracy. Previous works have suggested the difficulty of maintaining accuracy in presence of fatigue [33,34], whereas athletes have reported feeling poor accuracy in presence of mental fatigue [33]. Slower serves may facilitate this outcome.

Although this information should not be applied to professional tennis players (the main limitation of the study is that the sample was only made up of recreational players), these results provide important practical applications. Tennis coaches should take into account this information during training sessions and matches. First, tennis players should optimize the tennis serve during training and matches, even in the presence of fatigue [35]. Therefore, training strategies may be developed. For example, if we want to gain an advantage during the competition, it would be necessary to train the serves in the last moments of training (an example of ecological training) or synergizing mental and physical fatigue by laboratory-based tasks (i.e., HIIT and Stroop). Van Cutsem and Marcora [20] recommended the development of mental fatigue resistance in athletes, to maintain the performance in presence of mental fatigue, where the two options previously named are valid options for this purpose [36]. Secondly, this also implies the quantification of the mental load to control and manage the mental load during training sessions according to the coaches' objectives. Van Cutsem and Marcora have previously strongly recommended the use of training sessions without the highest demands of mental load to competitions. Finally, the use of mental fatigue recovery strategies during training may be important, due to players having to play more than one match per day in certain tournaments [37].

Also, we would like to highlight the importance of these results for sustainable lifestyles. In this study, we presented data on the effect of mental fatigue in sport manifestations. This information would help coaches and practitioners to better design training sessions as well as training periodization, where tennis is one of the main sports played by amateur athletes. Therefore, and especially in the actual context in where we live, where the population is exposed to large psychological demands, understanding how mental stress or mental load affects physical manifestations would allow the general population to prevent the limiting situations that negatively affect their physical activity and performance, helping to ensure the sustainability of their daily physical activity. Indeed, in a previous study [38], it has been suggested that mental fatigue may decrease the physical activity intentions by decreasing the motivation levels. As such, controlling the level of mental fatigue may also enhance the physical activity of amateur athletes, promoting a sustainable lifestyle.

In relation to future studies, it is also necessary to test whether this effect differs according to participants' gender [39] and the differences in the duration of these impairments because it could be necessary to implement specific recovery strategies [37]. It



would be also interesting to test the effects of cognitive fatigue separately on tennis service performance. Indeed, in future works it would be interesting to analyze both services (i.e., first and second service) in training and real matches, increasing the ecological validity of the data obtained.

## 5. Conclusions

The results of the present study suggest that both a single HIIT and a combination of HIIT with Stroop impair the speed of the tennis serve, without significant changes in the accuracy. These impairments were not associated with poor strength manifestations. Therefore, the slower serve reported after both protocols could be caused by a response to maintain accuracy in presence of fatigue, especially in the combined protocol.

**Author Contributions:** Conceptualization, J.P.F.-G. and V.J.C.-S.; data curation, J.D.-G. and M.Á.L.-G.; formal analysis, J.P.F.-G. and V.J.C.-S.; funding acquisition, J.P.F.-G.; investigation, J.P.F.-G., J.D.-G., and V.J.C.-S.; methodology, J.P.F.-G., J.D.-G., M.Á.L.-G., and V.J.C.-S.; project administration, J.P.F.-G.; resources, J.P.F.-G. and V.J.C.-S.; software, J.P.F.-G., J.D.-G., M.Á.L.-G., and V.J.C.-S.; supervision, J.P.F.-G. and V.J.C.-S.; validation, J.P.F.-G. and V.J.C.-S.; visualization, J.D.-G. and M.Á.L.-G.; writing—original draft, J.P.F.-G., J.D.-G., M.Á.L.-G., and V.J.C.-S.—review and editing, J.P.F.-G., J.D.-G., M.Á.L.-G., and V.J.C.-S. All authors have read and agreed to the published version of the manuscript.

**Funding:** This research was funded by the Ministry of Economy and Infrastructure of the Junta de Extremadura through the European Regional Development Fund, grant number GR18129.

**Institutional Review Board Statement:** The study was conducted according to the guidelines of the Declaration of Helsinki, and approved by the Ethics Committee of Extremadura´s University (CIPI/18/093).

**Informed Consent Statement:** Informed consent was obtained from all subjects involved in the study.

**Conflicts of Interest:** The authors declare no conflict of interest.

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
