# Peer review of "Effects of Combined HIIT and Stroop on Strength Manifestations, Serve Speed and Accuracy in Recreational Tennis Players"

_sustainability, doi:10.3390/su13147717_

Round 1
Reviewer 1 Report
Dear Authors,
The main aim of the submitted manuscript was to analyzed the effect of an High-Intensity Interval Training (HIIT) with and without cognitive load on serve speed and accuracy, spirometry, and strength manifestation.
This paper could be considered qualified to be published if the authors apply the modifications requested.
While it is a very interesting topic, some suggestions for improving the paper are provided below:
- Authors need to better define the meaning of “first service”. Therefore, the tests were not proposed during a real match. Indeed, the tests proposed are based on a very limited model of reality. What about the second service?
- Authors do not consider any specific difficulties or other aspects of performing the tennis service to minimally justify the comparison; on the other hand, the use of the descriptive knowledge obtained from the observed differences was not discussed in terms of applications for athletes and coaches. In short, the manuscript seems to be limited to a laboratory exercise to spatially and temporally describe the participants' movement. Authors need to better clarify the applications for athletes, physical trainers, physiotherapists, and coaches.
- Line 57: “Physical and cognitive fatigue” meanings have to be better described in the Introduction.
- Participants: Were there any exclusion criteria, e.g. injuries? – please add the information in the text. I would suggest to improve this section including the following sentence in the Methods “Exclusion criteria included injuries of any type that occur in the …month/years before the study.”
- Line 84: “An adaptation of the criteria established by Menayo et al [1]..”. Wich kind of adaptation? Authors need to justify this adaptation.
- Authors have to include information about errors, mistakes, faults during the service executions (i.e. in or out services).
Author Response
Thanks in advance for your suggestions,
we have included all the information in the doc.
Thanks!

Reviewer 2 Report
Thank you to the authors for your work on this manuscript. I find it quite novel and interesting. However, the methods section are not presented properly. My specific comments are presented below
Lines:
32: Please provide a reference
37 – 45; These lines do not present a similar idea. Please clarify them to make reading easier.
In my opinion, you have to better support using HIIT in this research.
Line 46: Please provide a reference after 1 sentence?
line 48: Please indicate what kind of sports performance (serve?) during racket sports have been impaired.
71: How did you calculate sample size? What was the inclusion and exclusion criteria? Any standardization about a diet and supplements and activity?
71:Why seven serves?
80: Did the participants performed it in a randomized order? Why didn't you check a cognitive load separately?
Line 86: Why sending a ball to a "failed" part of the court gives points? What has happened if somebody performed a "net service"
98; Any ref. for that kind of HIIT? any ref. of Stroop test? How was it conducted? What was the score?
100: Is this warm-up was conducted before HIIT or later? If later, any warm-up before HIIT?
115: What kind of horizontal jumps? when this warm-up was conducted? How did you measure the jump?
122: How did you measure isometric handstand (position, number of repetitions, etc?)
119: When did you measure it? You have to describe it more specifically.
Line 124: When did you measure body temperature?
Line 135: HIIT and cognitive load?
Table 1: units in [ ]; any unit of accuracy?
147-151: "valor" is not neccessery
Figure 2; Please indicate Speed [km/h ] in a y line
Table 2: It is possible that body temperature was lower than normally (36.6)?!
In the whole article: HIIT not HITT
Author Response

(The authors gave the same response as above.)

Reviewer 3 Report
The influence of various factors on tennis performance is a significant issue.
Serving in tennis success is one of the main determinants of a successful match. The identification of factors that influence serving in tennis is an important part of research areas in the category of sports science. This is the main problem of the manuscript.
This manuscript falls into the category of sports science. It is important for the scientific community and readers that articles be placed in the right categories.
Author Response

(The authors gave the same response as above.)

Round 2
Reviewer 2 Report
Thank you for the corrections. However, some of them must be improved:
71: Please provide a reference
84: You have to write about expected effect size (ES) (indicated on what study you have based ES) alpha= ?, the statistical power= ?, r = ?, ? group of participants, ? experimental conditions
115- the exclusion criteria have to presented in the section participants
116: I understand no using caffeine before 1 hour of the test but informing participants to not consume creatine 1 hour before the training has no sense because it not works in an acute way. This sentence means that it is possible that some participants used creatine and beta-alanine during the study period. It is true?
147: so it was a CMJ? What was the rest intervals between them? in fact, you have assessed muscle power, not strength.
157: What was the time of contractions and rest intervals?
160: What was the rest intervals between them?
161: What is BT? What was the rest intervals between tests? When ST was measured?
159: I'm not sure that in this reference we can find a procedure of IHS.
Table 1. km/h not Km/h
160-163: I'm not sure that it is a correct place for these sentences; they have to be at the end of this section
Table 2; it will be better to write instead of lower body strength a jumping assessment.. cm is not a unit of strength
Please add a limitation of this study - in case of using supplements during the study period (there is also no information about standardizations including diet, and activity the day before the test and the time of this study)
Author Response
Thanks for your time to improve the quality of this manuscript.
We have responded point-by-point.
Looking to further considerations of our manuscript.

Reviewer 3 Report
Accept in present form
Author Response
Thanks to the reviewer for the time to improve the quality of this manuscript.